# Oral Food Challenge

**DOI:** 10.3390/medicina55100651

**Published:** 2019-09-27

**Authors:** Mauro Calvani, Annamaria Bianchi, Chiara Reginelli, Martina Peresso, Alessia Testa

**Affiliations:** 1Operative Unit of Paediatrics, S. Camillo-Forlanini Hospital, Circonvallazione Gianicolense 87, 00152 Rome, Italy; mi5660@mclink.it (M.C.); annamaria.bianchi9@yahoo.it (A.B.); 2Department of Paediatrics, Sapienza University of Rome, Viale Del Policlinico 155, 00161 Roma, Italy; reginellichiara@gmail.com (C.R.); alessiatesta92@live.it (A.T.)

**Keywords:** oral food challenge, food allergy, food allergens, single-blind, placebo-controlled oral food challenge, double-blind, placebo-controlled oral food challenge

## Abstract

Oral food challenge (OFC) is the gold standard for diagnosis of IgE-mediated and non-IgE mediated food allergy. It is usually conducted to make diagnosis, to monitor for resolution of a food allergy, or to identify the threshold of responsiveness. Clinical history and lab tests have poor diagnostic accuracy and they are not sufficient to make a strict diagnosis of food allergy. Higher concentrations of food-specific IgE or larger allergy prick skin test wheal sizes correlate with an increased likelihood of a reaction upon ingestion. Several cut-off values, to make a diagnosis of some food allergies (e.g., milk, egg, peanut, etc.) without performing an OFC, have been suggested, but their use is still debated. The oral food challenge should be carried out by experienced physicians in a proper environment equipped for emergency, in order to carefully assess symptoms and signs and correctly manage any possible allergic reaction. This review does not intend to analyse comprehensively all the issues related to the diagnosis of food allergies, but to summarize some practical information on the OFC procedure, as reported in a recent issue by The Expert Review of Food Allergy Committee of Italian Society of Pediatric Allergy and Immunology (SIAIP).

## 1. Introduction

Food allergies have been increasing: A recent systematic review and meta-analysis report that lifetime self-reported prevalence of allergy to common foods in Europe ranged from 0.1 to 6.0% [1]. Diagnosis of a food allergy is not simple, and self-reported rates of food allergies are much higher than the true prevalence [2]. Clinical history and lab tests have poor diagnostic accuracy and they cannot make a certain diagnosis of a food allergy. Thus, the oral food challenge (OFC) is the gold standard for diagnosis of a food allergy [3]. This review does not intend to analyze comprehensively all the issues related to the diagnosis of food allergy, but to summarize some practical information on the OFC procedure, as reported in a recent issue by The Expert Review of Food Allergy Committee of Italian Society of Pediatric Allergy and Immunology (SIAIP) [4].

OFC comprises the oral administration of the suspected allergen in a controlled and standardized setting. It is a complex test, which requires large healthcare (physician, nurse, hospital facilities) and family (stress, fear) resources. OFC is useful to confirm or exclude the diagnosis of a food allergy (both for IgE-mediated and non-IgE mediated reactions), to assess the tolerability of a food in a child with a previous food allergy, or to identify the threshold of responsiveness [5]. Other indications for the oral food challenge are to test a particular food in sensitized patients who have never ingested that food or to test a cross-reactive food never introduced in the diet [5,6,7].

Children with a history of recent anaphylaxis (within 12 months) and detectable levels of IgE specific to a suspected food should be excluded from being tested with the oral food challenge [5]. Patients should also not be challenged if affected by atopic disease that might interfere with the assessment, diseases that might affect safety, or if they are taking drugs that might interfere with the assessment or affect safety [8].

## 2. Clinical History

An accurate collection of clinical history is essential to define when further diagnostic investigations are needed and how to implement them and interpret their results. Clinical history aims to identify food allergy cases by investigating symptoms, possible allergens, relationships between food ingestion and onset of symptoms, ingested dose, intercurrent diseases, potential cofactors or cross-reactivity, other allergies, the role of the suspected allergen in the diet, and possible effects of previous diets [9]. The goal of clinical history is also to identify the possible immunological mechanism underlying the food allergy. Some conditions may point to IgE-mediated allergic reactions such as signs of skin involvement (urticaria, angioedema, erythematous rash), respiratory features (rhino conjunctivitis, cough, dyspnea, or asthma) or gastrointestinal ones (oral itching, nausea, vomit, abdominal pain, and diarrhea), or even malaise and hypotension that occur within 2 h after ingestion of a probable allergen. Food Dependent Exercise Induced Anaphylaxis (FDEIA) is an exception as it arises after a greater temporal latency. The longest intervals between eating and onset of the symptoms were 3.5 h, while between the start of exercise and the onset of symptoms it was 50 min [10] Subjects affected by FDEIA are sensitized to the food responsible for anaphylaxis, even if specific IgE blood levels are lower than in other food allergies. Ingestion of the suspected food provokes clinical manifestations only when followed by physical exercise. At the same time, physical activity does not induce adverse reactions if not preceded by food ingestion. In allergen-specific FDEIA, the role of exercise (or other co-factors such as aspirin, alcohol, etc.) is crucial, because it prompts the development of clinical reactions to a food that is commonly eaten by the patient, without any clinical manifestation [11].

Delayed symptoms, however, especially those affecting the gastrointestinal tract, lead towards a non-IgE-mediated reaction or to a mixed IgE- and non-IgE-mediated reaction (Table 1).

## 3. Diagnostic Tests

The diagnosis of IgE-mediated food allergy relies on a compatible clinical history and on both the results of skin prick tests (SPTs) and the determination of serum-specific immunoglobulins E (sIgEs). Both tests have good sensitivity but low specificity, which means that they are often positive in non-allergic subjects [12]. It is well known that the greater the SPT wheal size or the sIgEs level, the greater the probability of showing allergic symptoms [13]. Thus, many authors have tried to determine a cut-off for wheal sizes or for sIgEs levels able to predict a positive challenge. Several systematic reviews [14,15], guidelines, and international consensus [5,8,16] have suggested the use of cut-off values to reach a diagnosis of some food allergies (e.g., milk, egg, peanut, etc.) without performing an OFC. However, the proposed cut-offs vary depending on various factors, such as type of food, clinical picture, the population enrolled in the study, age [17], cooking of food [18], the type of allergen used to perform SPTs (e.g., commercial extract, raw egg, or heated egg) [19] etc., and sometimes also taking into account these factors. The Food Allergy Committee of Italian Society of Pediatric Allergy and Immunology (SIAIP) recently published two systematic reviews on the predictive value of SPTs and specific IgEs for egg [14] and milk [20].

As for egg allergy, a total of 37 articles were included, and their methodological quality was evaluated according to the criteria given by the QUADAS-2 tool [21]. Despite the division into groups based on the degree of cooking, age, and the type of allergen used to perform the SPTs, proposed cut-offs, with the exception of cut-offs for raw egg, showed a large variability, especially for sIgEs.

The second systematic review on the predictive value of SPTs and specific IgEs for milk allergy included 31 papers and grouped them according to patients’ age, allergen type, and cooking degree of the milk used for the oral food challenge. Grouping studies has reduced the variability of the cut-offs proposed, but not substantially. However, in children <2 years, proposed cut-offs seem to be homogeneous enough. The studies with the highest methodological quality suggest a 95% positive predictive value (PPV) cut-off for sIgEs of 5 KUA/L [22] and a 98% specificity cut-off for prick-by-prick (PbP) with fresh milk of 8 mm [23].

The two systematic reviews concluded that, both for egg and milk allergies, no proposed cut-off can be used to definitely confirm a diagnosis of Cow’s Milk Allergy (CMA) or egg allergy. However, with these limits, above all in children <2 years, when sIgE against CM or egg, or when SPTs are above the cut-off indicated in Table 2 and Table 3, the real need for a diagnostic confirmation of CMA or egg allergy through an OFC should be carefully evaluated.

As regards peanut and nut allergies, a recent guideline for the diagnosis of peanut and tree nut allergies stated that an SPT of ≥8 mm or an sIgE ≥15 KU/L to peanut is highly predictive of clinical allergy. Cut-off values are not available for tree nuts. However, it is generally accepted that a cut-off SPT ≥8 mm for a specific tree nut is highly suggestive of clinical allergy [24].

**Table 2 medicina-55-00651-t002:** Cutoffs proposed for milk allergy by the methodologically best studies. From: Bianchi A et al., [15] and Cuomo B et al., modified [20].

	<2 years	Ref.	>2 years	Ref.
**Fresh Cow’s Milk (Skin Prick Test)**				
Cow’s milk (commercial extract)	6 mm (100% Sp) (LR 13.2)	Sporik, [25]	8 mm (100% Sp) (LR infinite)	Sporik, [25]
Fresh cow’s milk prick-by-prick (PbP)	8 mm (98% Sp) (LR 9.5)	Saarinen, [23]	9 mm (95% PPV)	Onesimo, [26]
α- Lactalbumin (commercial extract)	-	-	4.9 mm (95% PPV)	Onesimo, [26]
β–Lactoglobulin (commercial extract)	-	-	5.6 mm (95% PPV)	Onesimo, [26]
Casein (commercial extract)	-	-	4.3 mm (95% PPV)	Onesimo, [26]
**Baked Cow’s Milk (Skin Prick Test)**				
Cow’s milk (commercial extract)	-	-	15 mm (67% Sp) (LR 3.5)	Nowak-Wegrzyn, [27]
**Fresh Cow’s Milk (IgEs)**				
Cow’s milk	5 kU_A_/l (95% PPV) (LR 30)	Garcia-Ara, [22]	-	-
	3.5 kU_A_/l (98 Sp) (LR 12.5)	Saarinen, [23]	-	-
	4.18 kU_a_/l (100% PPV) (LR infinite)	Keslin, [28]	-	-

PPV, positive predictive value; Sp, specificity; LR, likelihood ratio; PbP, prick-by-prick.

**Table 3 medicina-55-00651-t003:** Cut-offs proposed for egg allergy by the methodologically best studies. From: Calvani M et al., [14].

	<2 years	Ref.	>2 years	Ref.
**Raw Egg (Skin Prick Test)**				
Raw egg (commercial extract)	SPT = 4 mm wheal (95% PPV) (LR 6.7)	Peters, [29]	SPT = 10 mm wheal (95% Sp) (LR 5.2)	Vazquez-Ortiz, [30]
Raw egg (PbP)	-	-	PbP = 14 mm wheal (95% PPV) n.d.	Mehl, [31]
Ovoalbumin (commercial extract)	-	-	SPT = 10 mm wheal (95% Sp) (LR 5.2)	Vazquez-Ortiz, [30]
Ovomucoid (commercial extract)	-	-	SPT = 8.5 mm (95% Sp) (LR 7.1)	Vazquez-Ortiz, [30]
**Heated Egg (Skin Prick Test)**				
Raw egg (commercial extract)	SPT = 5 mm wheal (100% Spec) (LR 7.3)	Sporik, [25]	SPT = 11 mm wheal (95% Sp) (LR 2.3)	Vazquez-Ortiz, [30]
Ovoalbumin (commercial extract)	-	-	SPT = 10.5 mm wheal (95% Sp) (LR 4.7)	Vazquez-Ortiz, [30]
Ovomucoid (commercial extract)	-	-	SPT = 13 mm wheal (95% Sp) (LR 2)	Vazquez-Ortiz, [30]
**Raw Egg (sIgE)**				
Raw egg	sIgE = 1.7 kUA/L (95% PPV) (LR 21.2)	Peters, [29]	sIgE = 3.6 kUA/L (95% PPV) (LR 11)	Vazquez-Ortiz, [30]
			sIgE = 6 kUA/L (95% PPV) (LR 6.4)	Sampson, [32]
			sIgE = 7.3 kUA/L (95% PPV) (LR 11.4)	Ando, [33]

PPV, positive predictive value; Sp, specificity; LR, likelihood ratio; PbP, prick-by-prick.

## 4. Novel Diagnostic Approach

Allergen component-resolved diagnostic testing (CRD), first proposed by Valenta about 20 years ago [34], is a method able to dose purified or recombinant allergens for the identification of specific molecules causing sensitization or clinical allergy. CRD can be performed either in single test formats or in a microarray, testing a range of over 100 purified allergens simultaneously. It has been demonstrated that CRD may increase allergy diagnosis accuracy, both in respiratory and in food allergy.

A recent EAACI Molecular Allergology User’s guide proposed that Molecular Diagnostics (MD) can improve total allergen IgE testing including where: (1) There are low abundant and/or labile food proteins in conventional allergy tests, (2) MD provides information on risk or severity-associated molecules, and (3) MD provides indicators of food-related cross-reactivity or (4) markers of genuine (species-specific) sensitization. Among the other main indicators of CRD, there are idiopathic anaphylaxis, delayed red meat anaphylaxis, wheat-dependent exercise-induced anaphylaxis, differentiated between high- versus low-risk molecules from foods giving rise to food-induced anaphylaxis (peanut, nuts, shrimp, etc.), baked egg or milk allergy (ovomucoid, casein), etc. [14,20,35,36,37].

On the contrary, it is of little use when there is a convincing history of IgE-mediated allergy and a positive SPT or sIgE to the relevant whole food allergen; this information is already sufficient to make a diagnosis [38].

Basophil activation tests (BATs) have been applied in the diagnosis of cow’s milk [39], egg [40], and peanut [41] allergies, showing higher specificity and more negative predictive value than SPTs and sIgEs, without losing sensitivity or positive predictive value. However, BATs are available only in a few laboratories, thus it is still limited for research purposes on food allergy [9].

## 5. How Can We Use Cut-Offs in Clinical Practice?

Two different kinds of cut-off values were proposed in the literature, both for SPTs and for sIgEs: Those based on a high positive predictive value (95% PPV) and those based on a high specificity (95% specificity). The first ones, being based on the predictive value, depend on the prevalence of an allergy in the studied population and are applicable in allergy centers where it is assumed that the prevalence of food allergy is similar to the one found in the studies providing the values [42]. On the contrary, the cut-off values based on 95% specificity do not change with the prevalence of the disease in the population and give us the chance to better select the children to test with OFC, given the high risk of a positive challenge. The positive predictive value (PPV) is the probability that a patient has a food allergy if the test is positive. However, these predictive values are dependent on the population prevalence and other variables (the food allergen in question, background history, age, sex, geographic location, ethnicity, and concomitant allergies). It is therefore not possible to easily apply predictive values across different populations and in different settings [9].

Apart from these specific cases, it has been suggested to use the likelihood ratio (LR), which does not depend upon the prevalence of the illness and offers a different diagnostic approach, applicable on single patients independently. The LR is realized by combining (1) the pre-test probability, which can be inferred from the clinical record of the patient with (2) the post-test probability, namely the diagnostic test result (e.g., SPT or Specific IgE for suspected food). The LR highlights the number of times that a given result can be more likely seen in a patient allergic to a specific food compared to a patient who tolerates that food. The higher the LR, the higher the probability of an allergy. If the LR >10, it makes the OFC useless. [9,43]. The LR can be obtained with a simple mathematic formula (sensibility/l- specificity) or by using Fagan nomogram, which simplifies the calculation to a percentage instead of an odd ratio.

Establishing the pre-test probability with precision is not always easy, but it can be determined with a certain approximation, knowing the medical history of the child. The more typical the medical history, the more probable it is that the clinical situation reported has been caused by a food allergy. 

## 6. When Can We Decide on Elimination Diet Without OFC?

In 2009, the French position paper on food allergy [16] stated that “OFC is not indicated in children with a clinical history suggestive of allergy and positive results in skin tests or specific IgE”. They defined *a suggestive clinical history* if cutaneous signs (eczema, rash, urticaria, angioedema), gastrointestinal signs (nausea, vomiting, diarrhea, abdominal pain), respiratory signs (rhino conjunctivitis, cough, respiratory distress, bronchospasm), and/or arterial hypotension occur shortly after ingesting the food (and positive IgE test results). Recently, Food Allergy Practice Parameter [3] suggests that “OFC is not prudent or necessary to make the diagnosis of IgE- mediated food allergy if the patient has an unequivocal and *convincing history* of clinical reactivity to a known food allergen and positive sIgE test results (SPT or sIgE measurement)”. However, in this document, a definition of an unequivocal and convincing history is lacking. In the past, Sampson defined *convincing history* as an immediate allergic reaction to a food within the previous 2 years that developed after an isolated ingestion of the suspected food and required emergency management by a physician. Whereas, a *suggestive history* consists of an immediate allergic reaction that occurred after the ingestion of a food (not necessarily in isolation) on one or more occasions but did not require emergency management by a physician or a convincing reaction that occurred more than 2 years before [32].

As no study has demonstrated the diagnostic efficacy of these or other definitions, the Expert Review of Food Allergy Committee of Italian Society of Pediatric Allergy and Immunology (SIAIP) [4] suggests a more conservative definition of convincing and suggestive histories of IgE-mediated food allergy (Box 1) and suggests that a diagnostic OFC can be deferred, due to the high probability of a reaction, where the patient presents:*A convincing history* in the presence of a specific SPT or IgE positive to the suspected food.*A suggestive history* in the presence of a specific SPT or IgE above the level of the suggested cut-off.

In all cases, the values of the wheal of the SPT, or the value of the specific IgE, do not constitute an absolute side effect to the OFC, nor to the initial diagnosis or to evaluate the development of tolerance later on.

In conclusion, specific test results can be linked with a specific probability of allergy. However, this probability is influenced by the clinical history, which determines the prevalence of an allergy in the specific patient that is being tested. It is therefore essential that the results of the investigations are interpreted in the context of the clinical history.

A simpler approach to integrate the clinical history and sIgE result using a table was recently proposed [44]. Starting from this approach, we have created a table which shows how it is possible, in broad terms in clinical practice, to evaluate the usefulness of an OFC, keeping in consideration both the clinical history and the results of the allergy blood or skin tests and the cut-offs. (Table 4).

## 7. OFC Procedures and Schedules

The OFC must be done when the patient has not eaten for at least 4 h (to anticipate immediate reactions) or for at least 12 h (in the case of non-immediate reactions).

Before the OFC, it is strongly recommended to [6,45]:Collect medical history, which can highlight the type and severity of the previous reactions, the diet followed, and potential interruptions.Collect the parents’ consent form and, age permitting, also of the patients; moreover, it is necessary to inform about the risks, benefits, outcomes, and potential limits of a positive and negative OFCInvestigate the possible interference of pharmaceutical drugs which can hide an allergic reaction (antihistamine type H1), make it more severe (antacids, antihistamines type H2, and proton-pump inhibitors), or interfere with the administration of pharmaceutical drugs necessary to treat a potential allergic reaction (beta-blockers).Have a thorough objective investigation, to ensure the child is de facto able to be receive the OFC and to obtain a comparative evaluation, pre- and post-challenge. Patients should not be challenged near treatment with systemic steroids (e.g., within 7–14 days) because disease rebound might confound the interpretation of the food challenge result.

### a) In IgE-Mediated Food Allergy

The oral food challenge (OFC) may be performed as an open, single-blind, or double-blind challenge [5]. In the open OFC, both the doctor and the patient (and the family members) are aware of the food being offered. It is the simplest OFC, the least expensive in terms of time and costs, and therefore the most frequently used. On the other hand, it has the disadvantage of being at risk of failing the challenge due to the psychogenic interference in the genesis of symptoms. For this reason, its reliability is greater in the first years of life and it is variable according to child’s personality and reported symptoms. 

Although a negative open OFC excludes a reaction to the food, a positive result with subjective symptoms, such as itching of the mouth area, sickness, or abdominal pain, is not conclusive for a firm diagnosis of a food allergy and should be confirmed by a blind OFC [5]. In the single-blind OFC, only the doctor knows the composition of the food that is being administered. This eliminates the patient’s psychogenic interference in the genesis of symptoms, but not the possible bias in the doctor’s interpretation of symptoms. In the double-blind placebo test (DBPCFC), neither the doctor nor the patient are aware of when the suspected food or placebo is being administered. This minimizes the doctor’s interpretation bias and the patient’s psychogenic interference. For this reason, the DBPCFC is considered the gold standard for the diagnosis of food allergies; however, due to its difficult implementation, it is only used for research purposes, for clinical purposes when an open or single-blind challenge result is ambiguous, or in selected cases where any psychogenic interference in the genesis of symptoms must be excluded. 

In conclusion, in the majority of cases in the first years of life or when there is a low risk of bias, connected/due to psychogenic factors, it can be performed openly. In all the other cases, it is preferable to perform a blind or double-blind placebo-controlled food challenge. 

Different schemes in terms of dosage, time of administration, and dose increase can be applied to perform an OFC in an IgE-mediated food allergy. One of the most used schedules is proposed by Practall [8]. It consists of seven growing doses of food with semi-logarithmic increase of protein: 3–10–30–100–300–1000–3000 mg. Table 5 shows the amount of proteins for different kinds of food (expressed as mg or ml). The starting dose should always be lower than the one which triggered the allergic reaction. It might be possible to adapt the starting dose to the one which was tolerated during the previous food challenge if the clinical history suggests a partial tolerance to the food in the patient.

An OFC with a very low starting dose (from 3–10 µg of protein) has been proposed for children with previous anaphylactic reactions [46]. The span of time between the doses usually varies from 15 to 30 min, according to the type of protocol employed in the studies [8]. In those in which it is necessary to identify the threshold dose in order to start the oral immunotherapy, longer time intermissions are required to better define the tolerance or the allergy at each step [47]. The OFC is followed by a period of observation of about 2 h after the last dose of the food.

Labial challenges or lip challenges (putting a small amount of allergenic food on the inner and outer border of lip) is often used in the UK pediatric allergy center as an alternative or initial step of OFC. Variation in how lip challenges is performed and interpreted limits the reproducibility and the validity of the test and a recent study showed poor sensitivity [48].

### a) In non-IgE-Mediated Food Allergy

The OFC in Food Protein Induced Enterocolitis Syndrome (PFIES) is always an open food challenge. A variety of protocols (FPIES) related to OFCs have been proposed. Although some protocols provide the entire dose in a single portion, a recent International Consensus suggests to administer the challenge food at a dose of 0.06 to 0.6 g, usually 0.3 g of the food protein per kilogram of body weight, in three equal doses over 30 min [49]. It is generally recommended not to exceed a total of 3 g of protein or 10 g of total food (100 mL of liquid) for an initial feeding and observe the patient for 4 to 6 h [5]. Lower starting doses, longer observation periods between doses, or both should be considered in patients with a history of severe reactions [50]. When a very low dose of food protein is administered and there is no reaction after 2 to 3 h of observation, some experts advocate that the patient could ingest a full age-appropriate serving of the food, followed by 4 h of observation [49].

### a) In Food-Induced Exercise-Induced Anaphylaxis

The diagnosis of FDEIA is challenging, because even in typical case histories, a reaction cannot always be reproduced during the challenge with the suspected food in combination with exercise. A review of 234 reported cases of FDEIA found that food exercise + challenges successfully induced symptoms in approximately two-thirds of cases [51]. FDEIA seems to be a partial state of tolerance to food, whose pathophysiological mechanisms are complex and unclear [52]. Physical activity is the best known and most common reported co-factor responsible for insurgence of symptoms. The diversity in the reported level of exercise is wide, ranging from walking or playing to strenuous exercise. In these patients, aspirin and alcohol should also be considered as co-factors (or augmenting co-factors) equally with exercise able to lower the threshold and simultaneously increase the severity of the anaphylactic reaction. A standardized model for food challenges with the addition of different co-factors has not yet been developed. Several challenge procedures have been published, which have in common premedication with acetylsalicylic acid, relatively large amounts of the suspect food, and strenuous exercise. [53,54].

Christensen recently proposed a standardized challenge method for wheat-dependent exercise-induced anaphylaxis (WDEIA) using gluten, which could be applied to other foods. Starting with an initial titrated food challenge without any co-factor(s), this was followed by a food challenge in combination with exercise (or another co-factor). The starting dose was 1/10 of the cumulated threshold based on the challenge without a co-factor, followed by 2/10, 3/10, and finally 4/10. After each dosage, a treadmill test was performed for 15 min at a submaximal workload adapted to physical abilities of the individual patient. The treadmill test was performed to a maximum of four times or until objective signs occurred [55].

## 8. Safety and Risk of an OFC

Before performing an OFC, all risks associated with the test and the therapy needed in case of reaction shall be highlighted to parents, in order to receive the correct written informed consent [56,57].

Systemic reaction rate (including lower respiratory and or laryngeal symptoms) differs depending on the inclusion criteria and the type of children admitted to the test [58]; the highest systemic reaction rate present in literature is 28% [59].

The severity of an adverse reaction cannot be always predicted. Several parameters have been used as indicators of development and severity of a potential adverse reaction during the OFC. 

The most relevant factors to be considered as risk indicators are [4]:(a)Possible underlying immunological mechanism (highest risk in IgE-mediated rather than non-IgE mediated, except the Food Protein Enterocolitis). IgE-mediated reactions include life-threatening anaphylaxis. Non IgE-mediated reactions induce above all gastrointestinal symptoms and could sometimes induce shock [60].(b)The kind of allergen (especially peanuts, nuts, and seeds). In a report of 32 fatalities due to ingestion of allergenic food, peanuts and tree nuts accounted for more than 90% of the cases [61].(c)The type of allergenic molecule (molecules with increasing resistance to hydrolysis, firing, or digestion are more dangerous). Ovoalbumin (OVA), the most abundant protein found in egg white, is quite sensitive to thermic denaturation, like other egg proteins, such as ovotransferrin and lysozyme [62] On the contrary, Ovomucoid (OVM) is relatively resistant to heat and is considered to be the dominant allergen in egg white [63].(d)Age of the child (the risk increases with increasing age). It is well known that food allergy reactions generally worsen with increasing age [64] and food anaphylaxis death increases as children get older [65]. In a retrospective multicenter study of 544 OFCs, the median age of children that developed anaphylaxis was significantly higher than that of children with multi-organ reactions or mild reactions (*p* = 0.03) [19].(e)Presence of asthma [3]. Food allergy-related fatal and near-fatal reactions in children have been reported to be caused by asthma. [66] In a report of 164 cases of food-induced anaphylaxis, a clinical history of asthma increased the risk of wheezing [odds ratio (OR) 2.2; 95% confidence interval (CI) 1.1–4.5] and respiratory arrest (OR 6.9; 95% CI 1.4–34.2). [67](f)Co-factors. The presence of some co-factors increases the risk of food allergy and affects the severity. Physical exercise, temperature, severe infections, pre-menstrual syndrome, emotional stress, the use of pharmaceutical drugs (FANS), and the ingestion of alcohol can enhance the development of some allergic reactions to a food [68].(g)Criteria used to consider positive OFC (if one continues OFC, despite the onset of objective symptoms there is a higher possibility of severe allergic reactions). Wainstein enrolled 89 children with peanut allergy in a prospective study. The challenge protocol provided for the challenges had to be continued beyond initial mild reactions. Among the 21 children who developed anaphylaxis, in only 3 cases was the initial reaction anaphylaxis. The author suggests that “if the challenges had been stopped and treated when the initial milder reactions occurred, it is possible that only 3/21 children would have developed anaphylaxis” [69]. Finally, since severe reactions might occur, the oral food challenge should be carried out by experienced physicians in a proper environment equipped for emergency, in order to correctly manage any possible allergic reaction. Intravenous access should be available before starting the OFC, especially if there is risk of anaphylaxis or other severe reactions, as in case of enterocolitis, or if there is any doubt about the possibility to place a cannula during an emergency [5]. It is also good practice to establish protocols to manage adverse reactions, including accurate posology for drugs.

## 9. How to Interpret the Results

There is currently no definitive agreement on the criteria for defining the positivity of a challenge in IgE-mediated food allergies. The decision to discontinue dosing is influenced by the patient’s characteristics (clinical history), specific study protocols, purpose for which is conducted (diagnosis of food allergy or acquisition of tolerance), or to identify the threshold of responsiveness.

The OFC can result:Positive, when clear objective signs of allergic reaction appear or repetitive (at least three times) or multiple subjective symptoms in several organ systems occur [6,8];Negative, when no symptoms occur; orNot conclusive (or conclusive only for partial tolerance) if the test is stopped before the total dose of food is ingested.

Interpretation of test results could be difficult in very young children, since some symptoms, both subjective and objective, may be hard to recognize. It is suggested to be aware of some signs that can prelude to more important symptoms such as behavioral changes (stop playing, appear quieter, stay in mother’s arms), refusing food, putting hands into the mouth, and scratching ears or neck [70]. Tickling or itching in the throat, nausea, abdominal pain, or discomfort are the most common symptoms preluding more severe reactions in older children.

The OFC is considered diagnostic of FPIES (i.e., positive) if the major criterion (repetitive emesis in the 1- to 4-h period after ingestion of the suspect food and the absence of classic IgE-mediated allergic skin or respiratory symptoms) is met with ≥2 minor criteria (lethargy, pallor, diarrhea 5–10 h after food ingestion, hypotension, hypothermia, increased neutrophil count of >1500 neutrophils above the baseline count [49].

## 10. Management of Allergic Reactions

In case of IgE-mediated allergic reaction, during the food challenge, the most important treatment purpose is to quickly stop the reaction, preventing the progression of its severity. Treatment with antihistamines, steroids, and infusion of 0.9% sodium chloride solution is usually applied in the presence of local or light–moderate systemic reactions, without cardiovascular and/or respiratory involvement [5]. In case of a systemic reaction development, IM in mid-outer thigh administration of adrenaline with a dose of 0.01 /kg up to a maximum of 0.5 mg is specified [5].

Rapid I.V. hydration (20 mL/kg normal saline bolus) is the first-line therapy for the severe acute reactions at large or during an FPIES positive OFC [49]. A single dose of intravenous methylprednisolone (1 mg/kg; maximum, 60–80 mg) is often used for severe reactions, based on the presumed T cell-mediated intestinal inflammation, although no studies support this recommendation [71]. The use of intravenous ondansetron (0.15–0.20 mg/kg) can be helpful in stopping emesis during FPIES-related OFCs, as reported by two small case series [72,73].

## 11. From OFC to Oral Immunotherapy: The Low Dose Challenge

The concept of diet has changed a lot in the last ten years, and this had an impact in the way to execute OFCs. Practice Parameters in 2006 stated that after the diagnosis of food allergy has been confirmed correctly “complete avoidance of the implicated food(s) is the only proven form of prophylactic management currently available” [74].

Since then various studies have changed the dogma that strict dietary avoidance facilitates development of tolerance [75]. In fact, various studies have shown that repeated ingestion of the food allergens, starting with small doses and gradually growing, is able to induce desensitization, which consists of raising the threshold of reactivity to foods, whilst receiving food [76]. It has also been shown that about 70% of children with cow’s milk or egg allergies are able to ingest muffin or waffle that contained milk or egg extensively heated, respectively [27,77].

Therefore, today, several guidelines suggest that, even if a strict dietary avoidance is the main food allergy therapy, the diet should be adapted to the tolerance of the individual child, being able to vary from a strict diet to the possibility of ingesting some proteins modified by cooking (such as extensively cooked milk and egg) [4,9]. Furthermore, since children seem to gradually develop tolerance to food, and it is a relief for families when their child can tolerate small amounts of these basic foods, even if larger doses may still cause symptoms, a low dose OFC was already proposed in 2006 [78]. Afterwards, several other authors have confirmed the effectiveness of this procedure [79] and, recently, Japanese Guidelines of Food Allergy (JGFA) have proposed that among the objectives of an OFC, there is also the identification of the amount of food that can be ingested without problems, in order to implement the “minimum diet of elimination”. They suggest that a positive OFC does not always imply the need for complete elimination of a food allergen from the diet. Small amounts of food may be allowed to be taken, if tolerated. In particular, JGFA suggest strict dietary avoidance only in children who have experienced severe reactions at very low doses during an OFC. On the contrary, in those who have tolerated small doses of the food but have had allergic reactions at higher doses, JGFA suggest to continue to ingest a safe dose of 1–10% of the threshold dose in the following days [80].

All of this has had an effect also on the way the test is conducted. As a matter of fact, while in the past the aim of the OFC was to reach a maximum dose which should approximate that of an age-appropriate portion of the food [81], nowadays this objective is always valid for the first OFC that the patient undergoes, when the aim is to demonstrate the allergy. After the first OFC, in the following tests, when the objective is to demonstrate the possible development of the tolerance, it has been suggested to perform a low dose challenge, especially in children with a persistent allergy. The aim of a low dose challenge is to try to introduce at least low doses of the food and keep them in the diet, in order to facilitate the development of the tolerance (or, at least, to improve the quality of life by reducing the necessity of a strict diet) [78], or in order to test if infants have had a minimal clinical tolerance that allows starting oral immunotherapy [82,83].

## 12. Conclusions

The diagnosis of food allergy is not simple because of its multiple clinical pictures and because diagnostic tests are not always sufficient to offer diagnostic certainty. However, the combination of history and diagnostic tests in some cases can provide sufficient diagnostic reliability to make the diagnosis of a food allergy without conducting an OFC. In all other cases, or if the certainty of diagnosis is sought, an OFC is required. Since severe reactions might occur, the oral food challenge should be carried out by experienced physicians in a proper environment, equipped for emergencies

## Figures and Tables

**Table 1 medicina-55-00651-t001:** Clinical symptoms of food allergy.

	Cutaneous	Ocular	Upper Respiratory	Lower Respiratory	Gastrointestinal	Cardiovascular	Central Nervous System	Others
**IgE-Mediated Food Allergy**								
	Flushing, pruritus urticaria, angioedema	Pruritus, conjunctival erythema, lacrimation, periorbital edema	Sneezing, rhinorrhea, congestion, hoarseness, cornage, tirage	Shortness of breath, wheeze, intercostal retractions, cough	Nausea, vomiting, diarrhea, pain, oral angioedema	Tachycardia, bradycardia, vertigo, hypotension, syncope	Hypo-reactivity, weeping, irritability, anxiety, drowsiness, loss of consciousness	Sense of impending doom Uterine contractions
**Non-IgE-Mediated Food Allergy**								
**AD ***	Rash eczema							
**FPIES ***	Pallor				Severe vomiting, diarrhea	Hypotension, shock	Lethargy	Hypothermia
**FPIAP ***					Bloody stools			Anemia
**Mixed IgE- and Non-IgE-Mediated**								
**EoE ***					Nausea, vomiting, retrosternal pain and/or burning, dysphagia, esophageal food impaction			Poor growth

* AD, atopic dermatitis; FPIES, food protein-induced enterocolitis syndrome; FPIAP, food protein-induced proctocolitis; EoE, eosinophilic esophagitis.

**Table 4 medicina-55-00651-t004:** Example: A 10-month child with a suspected allergic reaction after the ingestion of egg (heated). Modified from Stiefel and Roberts.

SPT Extract (mm) Specific IgE (KU/L)	Negative <3 <0.35	Low =3 =0.35	Intermediate >3 to 5 >0.35 to 1.7	High >5 >1.7
**High** Convincing history	Possible allergy (re-evaluate the anamnesis, or repeat SPT or conduct specific IgE, if they have not been conducted before); OFC	Probable allergy (evaluate if it is necessary to conduct OFC)	Very probable allergy (possible to avoid OFC)	Very probable allergy (possible to avoid OFC)
**Intermediate** Suggestive history	Few probable allergy (re-evaluate the anamnesis, repeat SPT, or conduct specific IgE); OFC	Possible allergy; OFC	Possible allergy; OFC	Allergy (possible to avoid OFC)
**Low** Little suggestive history	No allergy; no OFC	Few probable allergy; OFC	Possible allergy (re-evaluate the clinical history); OFC	Possible allergy (re-evaluate the clinical history); OFC

OFC, oral food challenge; SPT, skin prick test.

**Table 5 medicina-55-00651-t005:** Suggested dosage for principal foods, based on their protein content.

	*Dose 1*	*Dose 2*	*Dose 3*	*Dose 4*	*Dose 5*	*Dose 6*	*Dose 7*	*Total Dose*
Protein content of food	3 mg	10 mg	30 mg	100 mg	300 mg	1 g	3 g	4.4 g
Pasteurised cow’s milk *	0.1 mL	0.3 mL	0.9 mL	3 mL	9.1 mL	30.3 mL	90.9 mL	134.6 mL
Hen’s egg emulsified ^	24.19 mg (0.02 g)	80.64 mg (0.08 g)	241.9 mg (0.24 g)	806.4 mg (0.8 g)	2419.2mg (2.4 g)	8064.5 mg (8 g)	24,193.5 mg (24.1 g)	35.6 g
Semolina pasta dry ^#^	27.52 mg (0.02 g)	91.74 mg (0.09 g)	275.2 mg (0.27 g)	917.4 mg (0.91 g)	2752 mg (2.7 g)	9174 mg (9.75 g)	27,520 mg (27.5 g)	41.2 g
Cod raw mg ^$^	17.64 mg (0.01 g)	58.82 mg (0.05 g)	176.4 mg (0.17 g)	582.2 mg (0.58 g)	1764 mg (1.7 g)	5822 mg (5.8 g)	17,640 mg (17.6 g)	25.9 g
Shrimp raw ^&^	22.05 mg (0.02 g)	73.52 mg (0.07 g)	220.5 mg (0.22 g)	735.2 mg (0.73 g)	2205 mg (2.2 g)	7352 mg (7.3 g)	22,050 mg (22 gr)	32.5 g
Roasted peanut °	10.34 mg (0.01 g)	34.48 mg (0.03 g)	103.4 mg (0.1 g)	344.8 mg (0.34 g)	1.034 mg (1 g)	3.448 mg (3.4 g)	10.340 mg (10.3 g)	15.1 g
Hazelnut ^¶^	21.73 mg (0.02 g)	72.46 mg (0.07 g)	217.3 mg (0.21 g)	724.6 mg (0.72 g)	2.173 mg (2.1 g)	7.246 mg (7.2 g)	21.730 mg (21.7 g)	32 g
Walnut ~	28.57 mg (0.02 g)	95.23 mg (0.09 g)	285.7 mg (0.28 g)	952.3 mg (0.95 g)	2.857 mg (2.85 g)	9.523 mg (9.52 g)	28.570 mg (28.5 g)	42.2 g

Protein content http://nut.entecra.it/646/tabelle_di_composizione_degli_alimenti.html (da INRAN 2009): ^*^ protein content 3.3%, ˆ protein content 12.4%, ^#^ protein content 10.9%, ^$^ protein content 17%, ^&^ protein content 13.6%, **°** protein content 29%, ^**¶**^ protein content 13,8%, **~** protein content 10.5%.

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
