# Peer review of "Oral Food Challenge"

_medicina, 2019, doi:10.3390/medicina55100651_

Round 1
Reviewer 1 Report
The authors Calvani Mauro et al have submitted their review ‘Oral Food Challenge’ to the journal Medicina. The authors have well described the challenges in implementation and interpretation of OFC, which is the gold standard for diagnosis of food allergy. Furthermore, the authors have described factors which would affect the interpretation of results such as age of the child and ‘form’ of the suspect food. Management of allergic reactions has also been described.
The manuscript tries to present the OFC in a simple, easy to understand article. However, it has come out as an article which lacks scientific depth. It neither serves as a complete guide for OFC nor for diagnosis of food allergy. I strongly believe the article in its current form would not be widely cited and redundant. The manuscript is ridden with grammatical errors, lacks adequate references, certain sections are inexplicable. Therefore, the manuscript does not warrant acceptance in its current form.
Specific comments:
1. The abstract is very vague, and the authors have not clearly stated the purpose/aim of the review. The whole manuscript lacks direction and would not be cited widely due to its shallow coverage of a lot of sections.
2. Abstract : Format error in the last sentences.
3. Line 25: ‘of oral’ …
4. Line 32: Not clear if this sentence ‘There is also …. avoided.’ should be place here
5. Line 59: please explain if cutoffs of wheal sizes or sIgE is predictive of food allergy or OFC?
6. Line 64: There is a lack of scientific rigor when no attempt has been made to adequately discuss the factors affecting food allergy diagnosis. Table 2 and 3 has not been explained adequately.
7. The section “OFC procedure” does not provide expert guidelines for carrying out the actual procedure for OFC.
8. Line 88 : Error wrong Table number
9. Table 4: Formatting issues…not easy to follow as the rows and columns are not aligned.
10. Reference missing for line 121-123.
11. Line 127 – ‘receive the correct’
12. Line 129 – ‘criteria and the type’
13. Line 129 – please provide explanation for ‘systemic reaction rate’
14. Line 134 – Line 145 : Provide consistent advice for various risk factors
15. The section ‘From OFC to oral immunotherapy’ is ridden with grammatical errors.
16. It is inexplicable as to why the authors have abruptly end their manuscript.
Author Response
Reviewer 1
Comments and Suggestions for Authors
The authors Calvani Mauro et al have submitted their review ‘Oral Food Challenge’ to the journal Medicina. The authors have well described the challenges in implementation and interpretation of OFC, which is the gold standard for diagnosis of food allergy. Furthermore, the authors have described factors which would affect the interpretation of results such as age of the child and ‘form’ of the suspect food. Management of allergic reactions has also been described.
The manuscript tries to present the OFC in a simple, easy to understand article. However, it has come out as an article which lacks scientific depth. It neither serves as a complete guide for OFC nor for diagnosis of food allergy. I strongly believe the article in its current form would not be widely cited and redundant. The manuscript is ridden with grammatical errors, lacks adequate references, certain sections are inexplicable. Therefore, the manuscript does not warrant acceptance in its current form.
Response. I would like to thank the reviewer for the comments and suggestions which allowed us to improve the article substantially. Therefore we have revised the article deeply, adding numerous bibliographic references and several sections.
Specific comments:
REV 1. The abstract is very vague, and the authors have not clearly stated the purpose/aim of the review. The whole manuscript lacks direction and would not be cited widely due to its shallow coverage of a lot of sections.
Response. The purpose of the article is specified in a new sentence in the abstract. All the sections have been developed with more specific information and details.
REV 2. Abstract : Format error in the last sentences.
Response: Done. The format was corrected
Rev 3. Line 25: ‘of oral’ …
Response: Done. The sentence was corrected
Rev 4. Line 32: Not clear if this sentence ‘There is also …. avoided.’ should be place here
Response: Sentence was deleted.
Rev 5. Line 59: please explain if cutoffs of wheal sizes or sIgE is predictive of food allergy or OFC?
Response: the cut-offs are predictive of a positive OFC, since they are obtained from records of OFCs. However, because the positivity of an OFC is related to food allergies, the test is also predictive of food allergies. A new section about cut-offs has been added.
Rev 6. Line 64: There is a lack of scientific rigor when no attempt has been made to adequately discuss the factors affecting food allergy diagnosis. Table 2 and 3 has not been explained adequately.
Response: new sentences and a new section to explain table 2 and 3 were inserted
The section “OFC procedure” does not provide expert guidelines for carrying out the actual procedure for OFC.Response: The section was integrated with several sentences.
Line 88 : Error wrong Table numberResponse: The number was corrected
Table 4: Formatting issues…not easy to follow as the rows and columns are not aligned.Response: Our original table was aligned. I would like to ask the review to correct the formatting issue
Reference missing for line 121-123.Response: Reference was added
Line 127 – ‘receive the correct’Response: The word was corrected
Line 129 – ‘criteria and the type’Response: The word was corrected
Line 129 – please provide explanation for ‘systemic reaction rate’Response: Definition was included in the text
Line 134 – Line 145 : Provide consistent advice for various risk factorsResponse: New sentences about various risk factors were added.
The section ‘From OFC to oral immunotherapy’ is ridden with grammatical errors.Response: Grammatical errors have been corrected
It is inexplicable as to why the authors have abruptly end their manuscript.Response: A conclusion paragraph was added.
Reviewer 2 Report
Review Report
# 50 – later can be changed to delayed #68 – Instead of child, authors can consider using patient as OFC can be performed in patients #71 –risk of mistakes can be changed to risk of failing the challenge #88 – It should be table 4 #201 strict diet can be changed to strict dietary avoidance Line #212-#216 “In particular, JGFA suggest … threshold dose (36)“ these line should be reworded as the use of English is not correct. Line #217 similarly some authors .. This line is not clear to me. This is a review of the article on oral food challenges. Authors have nicely summarized the difference between open and blinded challenges Authors have nicely summarized food protein amounts for challenge Authors can describe OFC protocols for IgE and Non-IgE challenges under separate headings. Also, authors can better explain the protocols, like duration in between food ingestion and final monitoring after the last dose. Which medications to stop before the challenge, and other instructions to patient. Authors have summarized the cutoffs for milk and eggs only and have left out peanuts, and other allergens, which can be summarized in a table. The last part of Oral food immunotherapy can be omitted or if the authors want to keep it, it should be better explained. The last 6-7 lines need revision of english
Author Response
Reviewer 2
Comments and Suggestions for Authors
Review Report
1 # 50 – later can be changed to delayed
Response: Word was changed
2 #68 – Instead of child, authors can consider using patient as OFC can be performed in patients
Response: Word was changed
3 #71 –risk of mistakes can be changed to risk of failing the challenge
Response: Word was changed
4 #88 – It should be table 4
Response: Word was changed
5 #201 strict diet can be changed to strict dietary avoidance
Response: Word was changed
6 Line #212-#216 “In particular, JGFA suggest … threshold dose (36)“ these line should be reworded as the use of English is not correct.
Response: The sentence was reworded
7 Line #217 similarly some authors .. This line is not clear to me.
Response: New sentences were added
8 This is a review of the article on oral food challenges. Authors have nicely summarized the difference between open and blinded challenges Authors have nicely summarized food protein amounts for challenge Authors can describe OFC protocols for IgE and Non-IgE challenges under separate headings.
Response: we added separate headings and also a section for OFC in Food Induced Exercise Induced Anaphylaxis
9 Also, authors can better explain the protocols, like duration in between food ingestion and final monitoring after the last dose. Which medications to stop before the challenge, and other instructions to patient.
Response: Several sentences were added
10 Authors have summarized the cutoffs for milk and eggs only and have left out peanuts, and other allergens, which can be summarized in a table.
Response: A sentence on peanut and nut allergy was added.
11 The last part of Oral food immunotherapy can be omitted or if the authors want to keep it, it should be better explained.
Response: New sentences to explain better the link of the section of Oral immunotherapy with OFC were inserted.
12 The last 6-7 lines need revision of english
Response: Done
Reviewer 3 Report
I read the manuscript with great interest. The topic is up to date, interesting and well described.
I have some minor concerns:
DBPCFC is not the first choice, due to many factors - time consuming, can result in anaphylaxis. There are other methods of allergic diagnosis that are being researched to avoid challenges as much as possible. I think that a paragraph on component resolved diagnosis is necessary and definitely on BAT. The indications to DBCFC are not clear in the manuscript. Who would be a candidate? When can we decide on elimination diet without OFC? There are also trials on OFC followed with cofactors of allergic diseases (exercise, NSAIDS, alcohol) - it also should be commented, for example in LTP allergy or omega-5-gliadine allergy the result of challenge without cofactor can be negative and it does not exclude allergy.
Author Response
Reviewer 3
Comments and Suggestions for Authors
I read the manuscript with great interest. The topic is up to date, interesting and well described.
I have some minor concerns:
1) DBPCFC is not the first choice, due to many factors - time consuming, can result in anaphylaxis.
Response: I agree. A sentence was added
2) There are other methods of allergic diagnosis that are being researched to avoid challenges as much as possible. I think that a paragraph on component resolved diagnosis is necessary and definitely on BAT.
Response: A new section was added
3) The indications to DBCFC are not clear in the manuscript. Who would be a candidate?
Response: a sentence was added.
4) When can we decide on elimination diet without OFC?
Response: a new paragraph was added
5) There are also trials on OFC followed with cofactors of allergic diseases (exercise, NSAIDS, alcohol)
Response: We included new sentences on exercise induced anaphylaxis and cofactors.
6) It also should be commented, for example in LTP allergy or omega-5-gliadine allergy the result of challenge without cofactor can be negative and it does not exclude allergy.
Response: New sentences and a new section on FDEIA were added
Round 2
Reviewer 1 Report
I thank the authors for their remarkable effort and for making the necessary corrections and re-submitting the manuscript for review.
The authors have elaborated all the sections and provided in-depth discussion, guidelines and interpretation of OFC tests.
I am satisfied with the changes and agree to accept the manuscript for publication in its current form.
Reviewer 3 Report
Thank you for taking into account my suggestions. Manuscript improved.